# Evaluation of a Cyber Risk Assessment Approach for Cyber–Physical Systems: Maritime- and Energy-Use Cases

Ahmed Amro * and Vasileios Gkioulos

Department of Information Security and Communication Technology, Faculty of Information Technology and Electrical Engineering, Norwegian University of Science and Technology, 2815 Gjøvik, Norway; vasileios.gkioulos@ntnu.no
* Correspondence: ahmed.amro@ntnu.no

**Abstract:** In various domains such as energy, manufacturing, and maritime, cyber–physical systems (CPSs) have seen increased interest. Both academia and industry have focused on the cybersecurity aspects of such systems. The assessment of cyber risks in a CPS is a popular research area with many existing approaches that aim to suggest relevant methods and practices. However, few works have addressed the extensive and objective evaluation of the proposed approaches. In this paper, a standard-aligned evaluation methodology is presented and empirically conducted to evaluate a newly proposed cyber risk assessment approach for CPSs. The approach, which is called FMECA-ATT&CK is based on failure mode, effects and criticality analysis (FMECA) risk assessment process and enriched with the semantics and encoded knowledge in the Adversarial Tactics, Techniques, and Common Knowledge framework (ATT&CK). Several experts were involved in conducting two risk assessment processes, FMECA-ATT&CK and Bow-Tie, against two use cases in different application domains, particularly an autonomous passenger ship (APS) as a maritime-use case and a digital substation as an energy-use case. This allows for the evaluation of the approach based on a group of characteristics, namely, applicability, feasibility, accuracy, comprehensiveness, adaptability, scalability, and usability. The results highlight the positive utility of FMECA-ATT&CK in model-based, design-level, and component-level cyber risk assessment of CPSs with several identified directions for improvements. Moreover, the standard-aligned evaluation method and the evaluation characteristics have been demonstrated as enablers for the thorough evaluation of cyber risk assessment methods.

**Keywords:** cyber risk assessment; evaluation; cyber–physical systems; ATT&CK; FMECA; maritime; energy; autonomous passenger ship; digital substation



## 1. Introduction

Interest in cyber–physical systems (CPSs) has increased in recent years across different application domains such as maritime and energy. The maritime industry is undergoing a major transformation leading to changes in operations and technology [1]. As an example of this trend, this work is part of the "Autoferry" project [2] that aims to develop a ferry to transport passengers autonomously across the Trondheim canal. In our previous work [3], we classified the ferry as an autonomous passenger ship (APS).

Cyber attacks targeting the maritime domain are increasing both in number and severity [4]. Attacks of this type target all segments of the maritime infrastructure, including ships, ports, and shipping companies. The denial of service attack against the COSCO shipping company [5], stealing of confidential designs from Austal naval shipbuilders [6], and ransomware attack against Maersk [7] are notable and well-known examples. Arguably, attacks against ships are of relatively low complexity [8]. In real events, ships themselves have also been targets of attacks, and incidents involving their global positioning system (GPS) and communication technologies [9] indicate the feasibility of cyber attacks and potential impact.

Regarding the APS, it includes a wide range of information and communication technology (ICT), and industrial control systems (ICS) allowing it to be conceptualized as a CPS. The APS security risks may directly or indirectly endanger passengers' safety and adversely affect the operational environment. The ship can be steered into a collision with the surrounding environment or other ships if its remote or autonomous control capabilities are hijacked. To increase the trustworthiness, security, and resilience of integrated systems, risk management is considered for implementation in the APS architecture. As discussed in ISO 31010 [10] and ISO 27005 [11], risk management includes several processes, with risk assessment at the core. To ensure the safety of people and the systems themselves, the relationship between safety and security in the risk management of CPSs, such as autonomous ships, requires additional attention.

Moreover, surveyed risk assessment approaches in CPSs have been observed to rely heavily on experts' judgment which increases the required efforts for continuous risk assessment and management as well as having results that are heavily subject to bias [12]. In this direction, the authors of this article have previously proposed an approach for assessing the risks in cyber–physical systems [12]. The approach is based on failure mode, effects and criticality analysis (FMECA) risk assessment process and enriched with the semantics and encoded knowledge in the Adversarial Tactics, Techniques, and Common Knowledge framework (ATT&CK). We refer to this approach throughout this article as FMECA-ATT&CK. The approach reduces the need for expert judgment in several steps of the risk assessment leading to reduced efforts and impact of bias on judgment. In this article, further evaluation of FMECA-ATT&CK is carried and its results are presented. The evaluation relies on the engagement of a group of experts for conducting the risk assessment process using the FMECA-ATT&CK approach and another common approach, the Bow-Tie, against two different use cases in different application domains, one in maritime and another in energy. This allows for the evaluation of the approach based on a group of characteristics, namely, applicability, feasibility, accuracy, comprehensiveness, adaptability, scalability, and usability.

The contribution of this article can be summarized as follows:

- An evaluation of an open-source risk assessment process that is FMECA-ATT&CK supporting its development as a semi-automated cyber risk assessment tool for CPSs.
- Key characteristics for the evaluation of risk assessment methods. These characteristics can be utilized as a basis for comparison among existing and newly proposed methods for risk assessment.
- A standard-aligned methodology for the evaluation of risk assessment methods. The methodology allows for the evaluation according to a group of characteristics while reducing the impact of bias.

The remainder of this paper is structured as follows. In Section 2, background information regarding the relevant standards, methods, and use cases are provided. In Section 3, a group of related works is discussed to highlight the observed relevant methods and characteristics for evaluation. Then, the evaluation methodology is presented in Section 4. The evaluation of FMECA-ATT&CK is detailed in Section 5. The evaluation results are presented in Section 6. Then, reflections from conducting the evaluation including identified limitations and future directions are discussed in Section 7. Finally, concluding remarks are provided in Section 8.

## 2. Background

FMECA-ATT&CK has been developed as an approach for risk assessment. It has a defined set of inputs, and procedures, and produces an output. This allows it to be conceptualized as a system. Therefore, the evaluation is approached as a system analysis process following the ISO 15288:2015 system development standard [13]. Furthermore, the chosen system analysis method relies on experts' judgment through brainstorming, then techniques for eliciting expert views are utilized from the IEC 31010:2019 standard for risk assessment techniques [10]. Additionally, the applicability of FMECA-ATT&CK

for assessing risks in different use cases and application domains is among the targeted characteristics for evaluation. Therefore, two use cases are utilized to carry out the assessment procedure. One use case is a maritime-use case that is an APS while the other is from the energy domain that is a generic digital substation. In order to establish a reference for comparison, the evaluation includes the utilization of another well-established assessment process, the Bow-Tie, for evaluating the same use cases. In this section, several resources, approaches, and use cases are introduced to facilitate later discussion of the evaluation process of the FMECA-ATT&CK approach.

### 2.1. Standards, Methods and Approaches

The evaluation process is aimed to be aligned with the relevant standards and common approaches in the industry. The relevant standards are the IEC 31010:2019 [10], ISO 15288:2015 [13], and IEC 60812 [14]. Additionally, a commonly utilized method for risk assessment, the Bow-Tie, is utilized to provide a basis for a comparison with regards to FMECA-ATT&CK. Moreover, details regarding the use cases are presented hereafter.

### 2.1.1. IEC 31010:2019, ISO 15288:2015 and IEC 60812

The FMECA-ATT&CK approach has been developed based on the IEC 60812 FMECA standard [14]. The standard provides detailed steps for conducting a FMECA process including guiding criteria and suggested methods. The IEC 31010:2019 standard [10] was utilized for the identification of relevant risk analysis and assessment techniques to be adopted during the different steps in the FMECA process, such as the utilization of threat taxonomies as a threat identification method. Additionally, the IEC 31010:2019 standard was consulted during the evaluation of FMECA-ATT&CK regarding guidelines for eliciting expert opinions and judgment. Additionally, the system analysis process in the ISO 15288:2015 [13] standard for system development was consulted for the development of the evaluation methodology, particularly, the system analysis process. This highlights how aligned the FMECA-ATT&CK approach and its evaluation is with the relevant standards.

### 2.1.2. FMECA-ATT&CK

The Adversarial Tactics, Techniques, and Common Knowledge from MITRE, shortly known as the ATT&CK framework [15] is witnessing widespread adoption in both academia and the cybersecurity industry as a source of knowledge regarding adversarial tactics, techniques, and procedures (TTP). ATT&CK includes several technology domains such as enterprise information technology (IT) and the operational technology (OT) in industrial control systems (ICSs) and mobile technology making ATT&CK suitable in a wide range of use cases hosting a collection of these technologies. As opposed to other high-level models observed in the literature such as STRIDE [16] and the cyber Kill Chain [17], the ATT&CK framework presents a comprehensive and low-level abstraction of adversarial tactics and techniques. Additionally, the witnessed utilization of ATT&CK terminologies in threat reports [18] and cybersecurity testing frameworks, such as Caldera [19], highlights the utility of integrating the ATT&CK framework within different risk management processes starting with risk assessment.

Yet, ATT&CK is not a method for risk assessment. Therefore, a number of approaches have been considered in order to determine what method is most appropriate for risk assessment. We referred to the IEC 31010:2019 [10] standard for risk assessment techniques. The standard describes and compares the most commonly employed techniques in the different steps of risk assessment. We considered scope, time horizon, specialist expertise requirements, and the amount of effort required to apply risk assessment techniques. The scope of our risk assessment in a CPS includes components, equipment, and processes. In order to support continuous risk assessment and management as well as reduce the effect of biased assessment associated with expert judgment, the time horizon should be flexible. In addition, the amount of specialist expertise and effort needed should be at most moderate. We have chosen failure modes, effects, and criticality analysis (FMECA) [14] based on the

aforementioned criteria. Further, the standard emphasizes FMECA's application to all stages of the risk assessment process, which include identifying risk, assessing consequence, estimating likelihood, and evaluating risk.

Based on FMECA, FMECA-ATT&CK makes use of common knowledge encapsulated in the ATT&CK framework as shown in Figure 1. The components within the scope, their functions, and performance standards are defined. Then, the relevant failure modes are identified, and for this, the ATT&CK tactics are considered. Then, the existing detection methods are identified and their efficiency is estimated. Later, the impact of the consequences of failure is estimated based on five elements of impact, namely, operational, safety, financial, information, and staging (stage further attacks). The operational and staging impacts are estimated based on the centrality measures of the component after a graph of the system is modelled. The remaining elements are estimated based on expert judgment. Each failure mode is assigned a weighting of the expected impact elements. For instance, the collection tactic as a failure mode is only expected to cause information and staging consequences. In order to calculate the estimated failure mode for each component, each component is assigned criticality scores covering all five elements of impact. Afterwards, the possible failure mechanisms causing the failure modes are identified. For this, the ATT&CK techniques are utilized and their properties, such as relevant assets and platforms, are used to match them with the relevant components. After this, the likelihood of failure mechanisms is estimated based on the exploitability score in the common vulnerability scoring system (CVSS) which considers attack vector (AV), attack complexity (AC), privilege required (PR), and user interaction (UI). Later, the risk rating criteria are defined (e.g., based on value distribution). Finally, the relevant mitigation measures for each failure mechanism are defined. This is derived from the encoded knowledge in ATT&CK. When all the aforementioned information is collected, a risk priority number calculation and mitigation identification (RPNMI) algorithm is executed to calculate the risk of each failure mechanism and suggest the relevant mitigation measures. A detailed description of the FMECA-ATT&CK steps, tables, data types and sources of knowledge is presented in Appendix A. Additionally, the reader may refer to our original work [12] for more information regarding the risk assessment approach including a detailed comparison with other approaches ([12] §2.1).

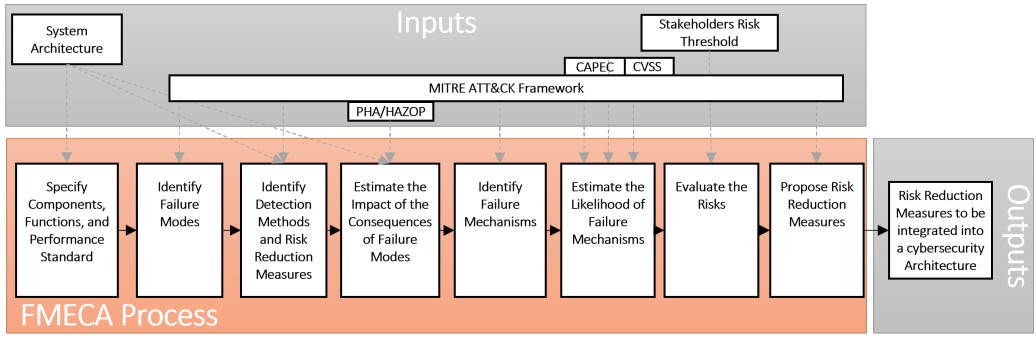

**Figure 1.** Steps of FMECA-ATT&CK with the knowledge sources (adapted from [12]).

### 2.1.3. Bow-Tie

The Bow-Tie approach allows for the assessment of cyber risks and the identification of barriers needed to control them without focusing on the likelihood. This aids in quick visualizations of the measures that are needed for implementation [20]. The Bow-Tie is a well-known method in the maritime sector and was found suitable for implementation in this paper to provide a basis for comparing the risk assessment results achieved through FMECA-ATT&CK and evaluating their soundness. The Bow-Tie method, as shown in Figure 2, begins with defining the scope of the target system for evaluation. This can be performed by interviewing system users, operators, and other stakeholders to answer scoping questions regarding the system components, and existing mitigation measures

to identify gaps. Then, threats and consequences are identified using any suitable threat modelling approach. This includes the identification of a top event, threat scenarios leading to it, and possible arising consequences. Then, the incident prevention and consequence reduction barriers are identified. Finally, the robustness and effectiveness of the barriers are considered to identify directions for improvement.

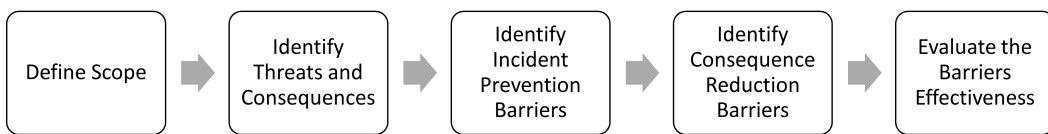

**Figure 2.** The Bow-Tie method (adapted from [20]).

### *2.2. Use Cases*

Two use cases are utilized to evaluate FMECA-ATT&CK, namely, an autonomous passenger ship (APS) and a generic digital substation (DS). The APS represents a use case from the maritime domain while the DS represents a use case from the energy domain.

#### 2.2.1. Autonomous Passenger Ship (APS)

The first use case is a prototype of an APS named milliAmper2. An overview of the use case description is depicted in Figure 3.

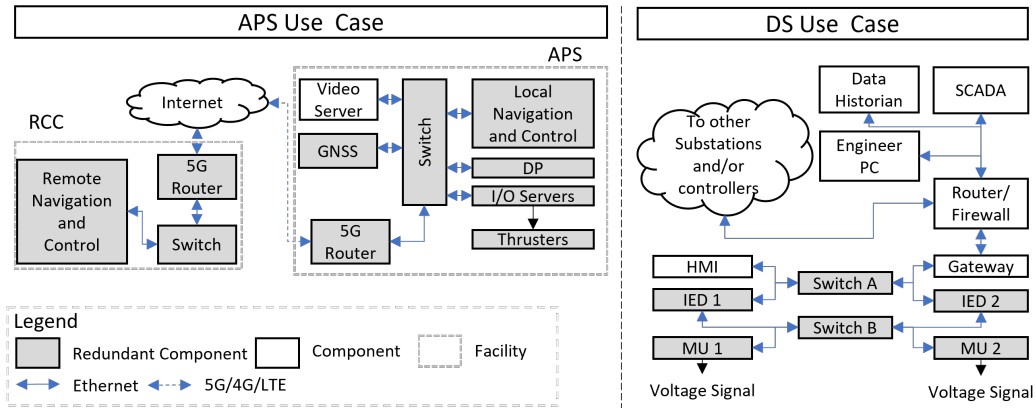

**Figure 3.** Overview of the APS and DS use cases.

The assessment scope was defined in one network among several networks of the milliAmper2 to reduce the assessment time and required efforts. However, information regarding redundant systems in other networks was utilized during the assessment for accurate risk estimation. In summary, the ferry includes several components such as a global navigation satellite system (GNSS) and video cameras as sensor data. These data are sent to a local navigation and control system to identify safe routes and then instruct the dynamic positioning (DP) system to control the thrusters through a group of I/O servers. The ferry is connected through a 5G network to a remote control centre (RCC) hosting a remote navigation and control system that can intervene in case of an unsafe situation. Further details about the APS architecture can be found in [21].

#### 2.2.2. Digital Substation (DS)

To evaluate FMECA-ATT&CK applicability in different use cases in the different application domains, another use case is needed. For this, we have identified the digital substation from the work of Khodabakhsh et al. [22] as a suitable use case. An overview of the use case description is depicted in Figure 3. The digital substation includes a supervisory control and data acquisition (SCADA) system with an engineering PC for monitoring and control. A data historian is hosted for storage. These components are connected through a router and a gateway to the lower devices, intelligent electronic devices (IED),

human–machine Interface (HMI), and merging units (MU) for monitoring and controlling the voltage signal.

## 3. Related Work

The area of cyber risk assessment in cyber–physical systems is rich with relevant literature. A wide range of risk assessment methods and approaches exist. However, limited works have been observed regarding a structured and systematic evaluation of such works.

Some works have been observed that targets the evaluation of risk assessment approaches. Tam [23] conducted a qualitative evaluation of the author's risk assessment framework. The author relied on expert judgment to measure the usability and applicability of the risk assessment framework using a survey. Abkowitz and Camp [24] investigated the applicability of the enterprise risk management (ERM) framework in marine transportation. The authors utilized a group of experts to implement the ERM framework against a case study including a marine transportation carrier. Grigoriadis et al. [25] engaged system stakeholders to evaluate a risk assessment tool regarding satisfaction of the stakeholders' security and privacy requirements as well as the feasibility of its application. The evaluation approach entails a demonstration of the tool and asking the participants to answer a questionnaire. Moreover, ref. [26] examined the feasibility of using the system theoretic process analysis (STPA) for risk analysis and quantitative risk modelling of autonomous ships. The author identified and assessed 35 risk analysis methods and found seven methods that can be used to enhance STPA for risk analysis of autonomous ships.

Additionally, several works proposing risk analysis and assessment approaches in CPS have been observed in the literature. The authors' evaluation of their contributions tend to include the utilization of certain use cases to demonstrate the applicability of their proposed approach (e.g., [27,28]). Some works have utilized other approaches to provide a ground for comparison (e.g., [23,26]).

Moreover, several guidelines and standards are available with relevant artefacts to evaluate risk assessment approaches. This includes the NIST assessment guidelines (NIST.SP.800-53Ar5) [29] and the ISO 31010:2019 risk assessment standard [10]. The NIST guidelines discuss several approaches for evaluation, namely, examine, interview, and test with different levels of rigour and scope ranging from basic to comprehensive. The ISO 31010 standard [10] suggests characteristics for comparison among risk assessment and analysis methods including application, scope, specialist expertise, and efforts to apply.

Lastly, our original work [12] proposing FMECA-ATT&CK as a risk assessment approach for CPS discussed the background and rationale. Among the original objectives is to include the applicability in different application domains utilizing information technology (IT) and operational technology (OT). Furthermore, the approach must be comprehensive in its consideration of risk elements. Additionally, the system must also reduce the need for expert judgment through employing the concept of curated knowledge to support the automation of some elements to allow a continuous risk assessment process. Moreover, the approach's adaptability to include additional components of risks has been demonstrated in the original work. Therefore, measuring the applicability of FMECA-ATT&CK, its comprehensiveness, and the accuracy of the results based on curated knowledge, automated elements and adaptability, was deemed necessary to assess the satisfaction of the original objectives and therefore these characteristics are targeted in this work.

## 4. Evaluation Methodology

An evaluation methodology for evaluating a risk assessment approach is proposed in this section. The approach under evaluation, FMECA-ATT&CK in this paper, is conceptualized as a system. Therefore, the evaluation is approached as a system analysis process following the ISO 15288:2015 [13] system development standard. This includes preparation, conducting, and managing the analysis results, as shown in Figure 4.

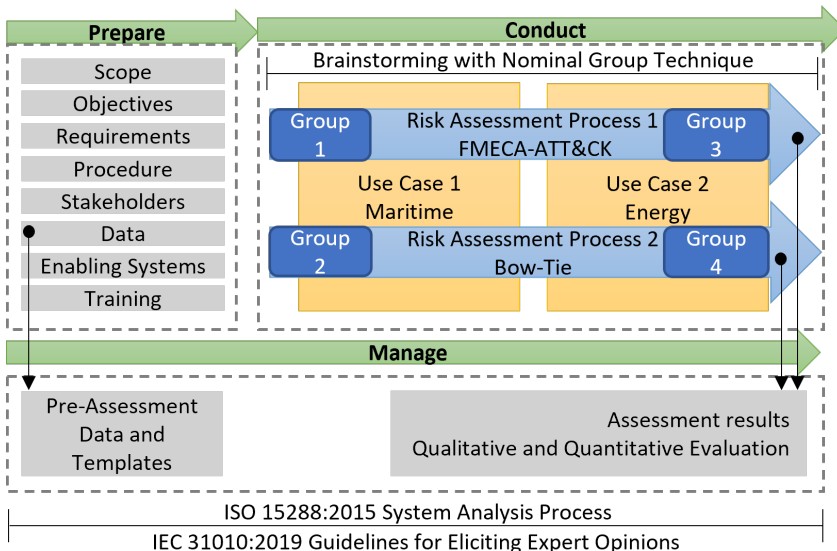

**Figure 4.** Evaluation methodology.

### 4.1. Preparing for the Evaluation

The preparation entails specifying the evaluation scope, objectives, and requirements. Then, the evaluation procedure is determined, the stakeholders are identified, and the required data and enabling systems are prepared. The scope is defined within the evaluation of a risk assessment method through application against several CPS use cases in different application domains. The evaluation objective is to evaluate the risk assessment method according to a group of characteristics, namely, applicability, feasibility, comprehensiveness, adaptability, scalability, usability, and accuracy. The characteristics were chosen based on what has been observed in the literature as well as the original motivations that led to the proposition of FMECA-ATT&CK. The evaluation characteristics are summarized in Table 1. Methods for measuring the characteristics are discussed in Section 5.1.2.

**Table 1.** Evaluation characteristics for the cyber risk assessment approaches.

| Characteristics | Objective | Related Work |
|---|---|---|
| Applicability | suitability for application in different use cases in different application domains. | [23,24] |
| Feasibility | the ability to implement the different steps in the approach. | [25,26] |
| Comprehensiveness | the extent to which different aspects of risks have been considered. Aspects of risks include, threats identification, likelihood and impact estimation, mitigation measures, etc. | [12] |
| Adaptability | The extent to which the missing aspects can be integrated to improve the method. | [12] |
| Scalability | The performance of the process in large and complex networks. | |
| Usability | The ability to follow and conduct the process with limited training/consultation. | [23] |
| Accuracy | The soundness of the results. | [23,26] |

Then, the analysis requirements are identified. The requirements are expected to be different according to each evaluation process. The following requirements are derived based on the method itself:

- Due to the reliance on expert judgment, measures for reducing bias in the assessment must be integrated in order to improve the assessment quality.
- Diversity in the use cases should be pursued to include various application and technology domains in order to measure applicability.
- Another common risk assessment method needs to be chosen that performs a similar function to the method that is subject to evaluation and provides categorically aligned results.

Additionally, an evaluation procedure should be defined. This includes applying the risk assessment that is subject to evaluation as well as another common and similar method against the same set of use cases. This is intended to provide a reference to compare the results. Moreover, the relevant stakeholders for the evaluation should be identified. The identification should consider their expertise in the application domain of the use cases. Finally, the data and enabling systems needed for the evaluation need to be prepared. This includes training the participants for the assessment.

### 4.2. Executing the Evaluation

The evaluation is proposed to be executed over several sessions spanning the different groups. As shown in Figure 5, the procedure is divided into three stages for each group, the first stage aims to run the assessment process step by step, describe to each participant the individual tasks, and address their questions. The participants should be given a sufficient period of time to provide their individual input. After receiving the participants' input, the results are evaluated to identify conflict areas and generate initial results based on consolidated inputs. Proposed consolidation rules can be found in Appendix B. In the third stage, the results are discussed in a group to reach a conclusion. Then, feedback from the participants applying the method under evaluation should be queried regarding their experience with the evaluated risk assessment approach utilizing a questionnaire.

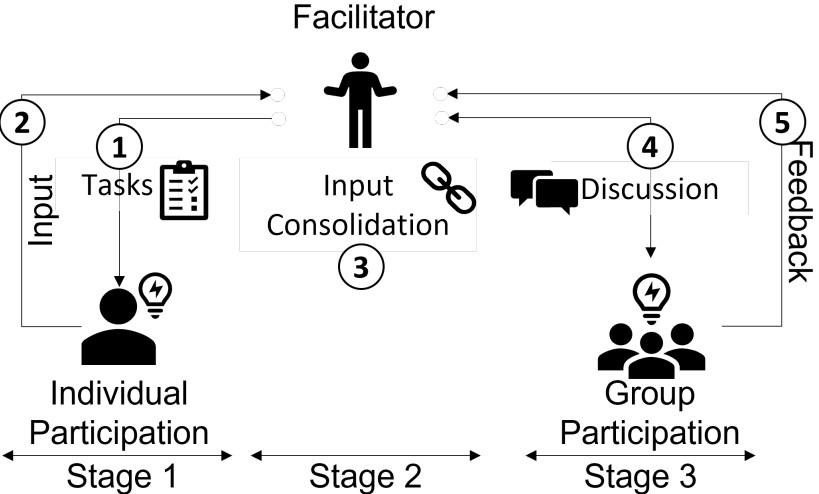

**Figure 5.** Execution Procedure.

### 4.3. Managing the Evaluation

All the prepared data for the evaluation, the participants' input, and the results should be maintained for future reference. This includes documents for conducting risk assessment processes by experts, the description of the use cases supporting their assessments, and the tools necessary to conduct the assessment processes, as well as their outputs.

## 5. FMECA-ATT&CK Evaluation

In this section, the evaluation process of FMECA-ATT&CK is presented. The evaluation is based on the methodology discussed in Section 4.

*5.1. Preparing for the Evaluation*

Preparation involves defining the scope, objectives, and requirements of the evaluation. Following this, the evaluation procedure is determined, stakeholders are identified, and the data and enabling systems are prepared.

5.1.1. Scope, Objectives and Requirements

The scope is constrained to the evaluation of FMECA-ATT&CK as a design-level cyber risk assessment approach in cyber–physical systems (CPSs). Two use cases of CPSs were chosen in different application domains, namely, an autonomous passenger ship (APS) or ferry representing the maritime domain and a generic digital substation (DS) representing the energy domain. The objectives include evaluating FMECA-ATT&CK according to the defined characteristics, namely, applicability, feasibility, comprehensiveness, adaptability, scalability, usability, and accuracy.

5.1.2. Evaluation Procedure

The evaluation procedure combines both qualitative and quantitative evaluation. All the characteristics are qualitatively evaluated after applying FMECA-ATT&CK in different use cases. The evaluation is based on experts' feedback through a questionnaire. A detailed evaluation criteria can be found in Appendix C. Additionally, to evaluate the applicability of FMECA-ATT&CK, we considered the utilization of two distinct use cases in different application domains as the target system of analysis. Additionally, usability is quantitatively measured by monitoring the experts' progression throughout the execution process. Moreover, the accuracy is qualitatively measured through a categorical comparison of the results obtained after the utilization of another commonly adopted risk analysis and assessment method carried out by different experts. The results of both methods are used as the basis for comparison to evaluate whether FMECA-ATT&CK is able to provide sound results.

Moreover, brainstorming to elicit experts' comments, concerns, and ideas regarding FMECA-ATT&CK was found to be a suitable approach for evaluation. However, since the chosen evaluation approach relies on expert judgment, the techniques for eliciting expert views are utilized from IEC 31010:2019 [10]. Additionally, since such views are subject to bias, the following measures for reducing bias were implemented:

- To reduce bias based on the bandwagon effect, the nominal group technique is implemented [30]. The bandwagon effect refers to the tendency of group ideas to converge rather than diverge. The nominal group technique has been found to generate more ideas than brainstorming alone [10].
- Group communication is hindered to avoid information bias.
- FMECA-ATT&CK itself implements measures to reduce bias through the utilization of metrics based on graph theory and data from the ATT&CK framework.
- Inputs from previous relevant risk assessment processes are avoided as much as possible. However, the utilization of some previous data was unavoidable. More details will be discussed later on when such a case occurred.

Then, the additional risk assessment process to be conducted was chosen to be the Bow-Tie method since it is a common approach in evaluating the risks in CPSs. Therefore, brainstorming with the nominal group technique while implementing FMECA-ATT&CK and Bow-Tie against different use cases was determined as the assessment procedure. Excel sheets were utilized as the medium for guiding the tasks and collecting the input from the experts. Noteworthy, each risk assessment process might require specific system-level information in a specific format. Therefore, a coherent state of the system description of both use cases must be maintained when applying the two processes to ensure a symmetric basis for evaluation. Furthermore, several groups each working on a different use case and applying a specific risk assessment process need to be formulated.

### 5.1.3. Identifying Stakeholders

The relevant stakeholders including the evaluation participants are identified and approached. In this direction, several subject matter experts (SMEs) were considered based on their experience in the use case application domain. As shown in Table 2, both academic and industrial SMEs were pursued with various experiences and backgrounds to improve the quality of the evaluation process.

**Table 2.** Experts roles and background.

| Group | Assessment Process | Participants | |
| | | Current Roles | Background and Previous Roles |
| --- | --- | --- | --- |
| 1 | FMECA-ATT&CK on APS | PhD candidate in maritime cybersecurity | Working experience on off-shore vessels |
| | | Researcher in cybersecurity | Maritime, energy, and CPS cybersecurity |
| | | PhD candidate in maritime cybersecurity | Seafarer (AB apprentice, AB, Deck Cadet, Junior Officer) and FMEA Auditor of DP systems |
| 2 | Bow-Tie on APS | Cybersecurity Consultant | IT/OT cybersecurity |
| | | Cybersecurity Consultant | IT/OT cybersecurity |
| 3 | FMECA-ATT&CK on DS | PhD Candidate in CPS cybersecurity | Cybersecurity in the smart grid |
| | | Postdoctoral researcher | Postdoctoral researcher in smart grid communication and security simulation |
| | | Researcher in cybersecurity and privacy | Cybersecurity in the smart grid and IoT privacy |
| 4 | Bow-Tie on DS | Industrial PhD/Cybersecurity Engineer | Cybersecurity in the smart grid |
| | | Industrial PhD/Senior adviser information security | Working with SCADA/OT—systems in the electricity sector for over 30 years |

### 5.1.4. Preparing Data, Enabling Systems and Training for the Assessment

The required data and enabling systems for the assessment were identified after a detailed study of the two risk assessment processes, namely, FMECA-ATT&CK and Bow-Tie. The common starting point for both is to define the analysis scope and this includes the targeted system for evaluation (i.e., use case). Descriptions of the two use cases to be targeted by both assessment processes were formulated. Several views of each use case architecture were prepared and made ready for analysis (Sections 2.2.1 and 2.2.2). The following steps are different for each risk assessment process. The steps for conducting FMECA-ATT&CK were followed based on our original article where it was proposed [12], while the steps for conducting Bow-Tie were based on the class guidelines published by De Norsk Veritas (DNV) [20].

When preparing the data required to conduct FMECA-ATT&CK, some data do not rely on expert judgment as it is extracted from the continuously updated ATT&CK framework, others required modelling the use cases in graphs to calculate the centrality metrics, while others were identified to require input from the experts.

The DNV class guidelines [20] were utilized for preparing the data required for conducting Bow-Tie. This entails answering a list of questions to help guide the experts in identifying the scope of the analysis and assessing the risks.

Finally, to facilitate a productive risk assessment process with the limited time the experts were willing to provide, the participants received a briefing and training on the assessment procedure and were provided with the required preliminary data using a combination of meetings and email communications.

*5.2. Executing the Evaluation*

The evaluation was executed over several sessions spanning the different groups following the procedure depicted in Figure 5. During the first stage, the assessment processes, namely Bow-Tie and FMECA-ATT&CK, were conducted step by step, describing to each participant the individual tasks, and addressing their questions. A sufficient period of time was provided so that participants could provide their individual input. On the basis of consolidated inputs, the results were evaluated to identify conflict areas and generate initial results. To reach a conclusion, the results were discussed in a group stage. Participants applying FMECA-ATT&CK were then surveyed regarding their experience with the risk assessment approach utilizing a questionnaire. An exception to that procedure occurred with the fourth group as we were forced to accommodate the participants' time constraints by running the first stage as a group by facilitating and stimulating discussion among the group participants and receiving their input for the assessment. After consolidating their input, the results were sent to the participants to receive their confirmation on the final assessment results.

5.2.1. Delivering Tasks and Receiving Input from Experts

The tasks for conducting FMECA-ATT&CK were compiled in an Excel worksheet including all the steps, tables to provide guiding notes, and extra room to receive detailed comments. The utilized template for each use case can be found in the authors' public repository (https://github.com/ahmed-amro/FMECA-ATT-CK-Evaluation) (accessed on 28 February 2023). On the other hand, the tasks for conducting Bow-Tie were communicated to the relevant groups in meetings. They were provided with the data prepared for the assessment. They were requested to deliver input to draft Bow-Tie diagrams and highlight the top threats, and mitigation measures required. After a sufficient period, the experts provided their answers.

5.2.2. Evaluating the Results

After receiving the experts' input. A consolidation process was executed to produce the risk assessment results (see Appendix B). The inputs for the FMECA-ATT&CK process were fed into a semi-automated tool to generate the results. On the other hand, the inputs for the Bow-Tie process were utilized to draft Bow-Tie diagrams. Finally, the results were presented to the experts, and a discussion was opened to reach a conclusion. Both sets of outputs were then compared to evaluate the soundness of the FMECA-ATT&CK results.

Another input was received from the groups applying the FMECA-ATT&CK process. This includes quantitative evaluation in the form of a rating of the process based on the characteristics specified earlier (Section 5.1.1) as well as qualitative evaluation in the form of comments on the process.

*5.3. Managing the Evaluation*

All the prepared data for the evaluation, the participants' input, and results are maintained in a public repository for future reference (https://github.com/ahmed-amro/FMECA-ATT-CK-Evaluation) (accessed on 28 February 2023). This includes the following:

- The utilized template for receiving experts' input for each use case when conducting FMECA-ATT&CK.
- The scoping questions and the prepared answers for the Bow-Tie process.
- The FMECA-ATT&CK scripts, inputs, and outputs.
- The generated Bow-Tie diagrams.

Moreover, this paper constitutes a report of the executed evaluation with the lessons learned.

**6. Evaluation Results**

The results of the evaluation are presented in this section. The evaluation relied on three types of input, namely, categorical comparison between the results of the risk

assessment processes, namely, FMECA-ATT&CK and Bow-Tie, experts' feedback through questionnaires, and experts' comments.

### 6.1. Risk Assessment Results

The results from the risk assessment processes conducted by the four groups have been collected, categorized, and compared. The categorization is based on the identified risks and suggested controls. More details in this regard are presented hereafter.

### 6.1.1. Top Risks

The Bow-Tie and FMECA-ATT&CK methods are categorically compatible. Threats, consequences, and mitigations in Bow-Tie can be mapped to techniques, tactics, and mitigations in FMECA-ATT&CK, respectively. This allows for a comparison of the results of the two methods and consequently provides evidence regarding the soundness of the results obtained through the application of FMECA-ATT&CK. The top risks identified through Bow-Tie and their relevant identified techniques in FMECA-ATT&CK and their corresponding risks are presented in Table 3. The results suggest that FMECA-ATT&CK can identify similar risks to Bow-Tie with more granularity-defined atomic techniques. The highest risks identified through FMECA-ATT&CK are all identified through Bow-Tie while several threats identified through Bow-Tie were rendered low risks. The rationale for these discrepancies is the consideration of existing mitigation measures. FMECA-ATT&CK does consider the existing mitigation measures in the risk calculation while the experts applying Bow-Tie appear to have either dropped them from their considerations or found them inefficient. Still, some threats identified through Bow-Tie (e.g., employees wrongdoing) are not supported by the current version of FMECA-ATT&CK which only considers adversarial threats. Due to the technical nature of some attack techniques, we provided the ATT&CK ID for the reader to refer to them in the ATT&CK framework repository. https://attack.mitre.org/ (accessed on 28 February 2023).

**Table 3.** The relations between the top risks identified through Bow-Tie and FMECA-ATT&CK.

| Bowtie Threats | FMECA-ATT&CK Techniques (ATT&CK ID) | FMECA-ATT&CK Risk |
|---|---|---|
| | | APS Use Case |
| Valid Accounts Stolen from a Student | Valid Accounts (T0859/T1078) | Low risk due to the inclusion of many relevant mitigation methods (e.g., access management) |
| Remote Desktop Protocol (RDP) | Remote Desktop Protocol (T1021.001) | Low risk due to the inclusion of many relevant mitigation methods (e.g., network segmentation) |
| Compromises Hosts | High-level threat. Relevant techniques: Drive-by Compromise (T1189), Compromise Client Software (T1554) | Both relevant techniques have a low risk either due to the inclusion of several relevant mitigations methods or low estimated impact and likelihood (e.g., update software) |
| Internal Spear phishing | Internal Spear phishing (T1534) | Low due to low estimated likelihood |
| Malicious Software | Malicious File (T1204.002) | Low risk either due to the inclusion of many relevant mitigation methods, and low estimated likelihood (e.g., execution prevention) |
| Compromised Credentials | High-level threat. Relevant techniques: Valid Accounts (T0859/T1078) Default Credentials (T0812) | Low risk due to the inclusion of many relevant mitigation methods (e.g., access management) |
| Single 4G/5G link | Outside the scope of FMECA-ATT&CK which only considers adversarial threats. | Although no techniques are identified for this specific threat, FMECA-ATT&CK does consider the existing redundant services to calculate the detectability (risk reduction degree). |
| Malicious Remote Access Tools | Exploitation of Remote Services (T1210) | Low risk due to the inclusion of many relevant mitigation methods (e.g., update software) |
| Legitimate Credentials with Native Network and Operating System Tools | Remote Services (T1021) | High risk for some components due to high likelihood, impact, and lack of existing relevant mitigation measures |
| Remote Services | | |
| Commonly used port (RDP, SMB, SSH, etc.) | Commonly Used Port (T0885) | Lw risk due to the inclusion of many relevant mitigation methods (e.g., network segmentation) |
| Repetitive Change of the I/O point values at the Control computer | Brute Force I/O (T0806) | Low risk due to the inclusion of many relevant mitigation methods (e.g., network segmentation) |



**Table 3.** *Cont.*

| Bowtie Threats | FMECA-ATT&CK Techniques (ATT&CK ID) | FMECA-ATT&CK Risk |
|---|---|---|
| DS Use Case | | |
| Supply Chain Compromise | Supply Chain Compromise (T1195) | High risk for some components due to high likelihood, impact, and lack of existing relevant mitigation measures |
| Wrongdoing by Employees | Outside the scope of FMECA-ATT&CK which only considers adversarial threats. | N/A |
| External Environmental Threats | | |
| Gaining Access to the System | 20 techniques in the "Initial Access" Tactic (TA0001 and TA0108). | High risk for some components due to high likelihood, impact, and lack of existing relevant mitigation measures |
| Ransomware | Data Encrypted for Impact (T1486) | Low risk due to low likelihood and existing relevant mitigation measures |
| Malware Injection | Malicious File (T1204.002) | Low risk due to low impact and existing relevant mitigation measures |
| Rouge Devices | Rogue Master (T0848) | Low risk due to low likelihood and existing relevant mitigation measures |

Another categorical view is the desired attacker objectives or expected consequences in the evaluated systems. The identified consequences through Bow-Tie and their corresponding tactics (i.e., objectives) identified by FMECA-ATT&CK are presented in Table 4. The results suggest an alignment of the identified possible consequences in both use cases. The results of Bow-Tie cover all the high risk objectives identified by FMECA-ATT&CK. Some consequences from Bow-Tie are rendered medium to low risks in FMECA-ATT&CK due to existing risk mitigation measures. Additionally, FMECA-ATT&CK identified additional objectives which Bow-Tie did not. This includes privilege escalation, exfiltration, credential access, and others.

**Table 4.** The relations between the top consequences/objectives identified through Bow-Tie and FMECA-ATT&CK.

| Bow-Tie Consequences | FMECA-ATT&CK Tactics | C | H | M | L |
|---|---|---|---|---|---|
| APS Use Case | | | | | |
| Malicious actions with logged in user privileges | Initial Access | 0 | 0 | 7 | 313 |
| Attackers with more information about the system | Discovery | 0 | 0 | 0 | 398 |
| Loss of view and control of the ferry from RCC | Impact | 0 | 15 | 78 | 301 |
| Attackers propagate and move freely within the network | Lateral movement | 0 | 3 | 18 | 239 |
| Malicious control over compromised hosts | Command and Control | 0 | 54 | 206 | 208 |
| An undesired system state or action is reached | Impair Process Control | 0 | 0 | 0 | 51 |
| DS Use Case | | | | | |
| Covert access to the system | Command and Control | 0 | 53 | 69 | 239 |
| Gaining physical access to the system | Initial Access | 0 | 4 | 2 | 40 |
| Losing trust of the system | Impact * | | | | |
| Credibility and societal trust | | | | | |
| Human harm | Impact | 0 | 6 | 21 | 263 |
| Reputation damage | | | | | |
| Loss of revenue | | | | | |
| Render system non-functional | Impair Process Control | 0 | 0 | 0 | 35 |

C: Critical, H: High, M: Medium, L: Low. * Trust is not an element of impact estimation; however, losing trust in the system is perceived by the assessors because of the functional impact.

### 6.1.2. Suggested Risk Controls

The identification of required risk mitigation measures or controls is a main objective of FMECA-ATT&CK. It was originally proposed as an instrument for the identification of risks and the proposition of the required controls to be considered in a subsequent process which includes the development of an architecture for cyber risk management. Table 5 depicts the controls suggested by Bow-Tie and the corresponding controls suggested by FMECA-ATT&CK. The controls already included in the use case are highlighted. Additionally, the number of identified high and medium risks for which the corresponding controls are suggested are presented as well. This suggests a certain priority of certain controls over others. The results suggest that the controls suggested by Bow-Tie and FMECA-ATT&CK are comparable. Most

of the controls proposed by Bow-Tie are also identified by FMECA-ATT&CK to address high to medium risks in both use cases. Some controls are proposed in FMECA-ATT&CK but not in Bow-Tie such as data backups for the APS use case. On the other hand, some controls suggested through Bow-Tie are not supported by FMECA-ATT&CK due to the scope. FMECA-ATT&CK only addresses controls that are relevant to the system's components.

**Table 5.** The relations between the top controls identified through Bow-Tie and FMECA-ATT&CK.

| Bow-Tie Mitigations | FMECA-ATT&CK Mitigations | Already Included | Suggested for | |
|---|---|---|---|---|
| | | | H | M |
| **APS Use Case** | | | | |
| Audit the Remote Desktop Users group membership regularly. | Audit | Yes | 0 | 0 |
| Remove unnecessary accounts and groups from Remote Desktop Users groups. | Use Account Management | Limited ** | 3 | 34 |
| Secure remote access to internal PC's and PLC's | Access Management, Account Use Policies, Authorization Enforcement, Human User Authentication, Password Policies, Software Process and Device Authentication, User Account Management, Multi-factor Authentication | Partially * | 3 | 4 |
| Secure portable media | Limit Hardware Installation, Antivirus/ Anti-malware, Behaviour Prevention on Endpoint, Execution Prevention, Exploit Protection | Limited ** | 3 | 90 |
| Clean support computers | Antivirus/Anti-malware | Limited ** | 0 | 6 |
| Regular patching, minimal applications, AV scan etc. for the jump server | Security Updates, Update Software, Use Recent OS Version, Vulnerability Scanning | Partially * | 0 | 8 |
| Email Gateways | Not supported | | | |
| Redundancy of 4G/5G Service | Redundancy of Service | Yes | 0 | 0 |
| Network Segmentation | Network Segmentation, Limit Access to Resource Over Network | Yes | 0 | 3 |
| Strict Access Control and Management of Change (MoC) with proper Validation | Not supported | | | |
| Firewalls | Filter Network Traffic, Limit Access to Resource Over Network, Network Allow lists, SSL/TLS Inspection | Limited ** | 33 | 126 |
| Intrusion Detection Systems | Behaviour Prevention on Endpoint, Network Intrusion Prevention | Very Limited | 48 | 195 |
| Not Discussed | Data Backup | Very Limited | 3 | 27 |
| **DS Use Case** | | | | |
| Following Standards and Routines | Not supported | No | | |
| Asset Management | Not supported | No | | |
| Security Testing | Deploy Compromised Device Detection Method, Vulnerability Scanning | No | 4 | 10 |
| Redundancy and Resilience | Redundancy of Service | Partially * | 0 | 0 |
| Access Control and Management | Access Management, Account Use Policies, Authorization Enforcement, Human User Authentication, Password Policies, Software Process and Device Authentication, User Account Management, Multi-factor Authentication, User Account Control | Partially * | 0 | 5 |
| Segmentation | Network Segmentation, Limit Access to Resource Over Network | Yes | 0 | 14 |
| Certification | Not supported | No | | |
| Awareness, Competence, and Skills Building | User Guidance, User Training, Application Developer Guidance | Yes | 0 | 0 |
| Business Continuity Plan (BCP) | Not supported | No | | |
| Recovery Capability | Data Backup, Remote Data Storage | Yes | 0 | 1 |
| Isolation Mode | Not supported | No | | |
| Incident Response, Detection, and Logging | Audit, Behaviour Prevention on Endpoint, Deploy Compromised Device Detection Method, Exploit Protection, SSL/TLS Inspection, Network Intrusion Prevention | Very Limited | 59 | 75 |
| Not Discussed | Filter Network Traffic | Partially * | 12 | 23 |
| | Update Software | Limited ** | 4 | 14 |
| | Execution Prevention | No | 3 | 3 |
| | Encrypt Sensitive Information | No | 1 | 4 |

H: High Risks, M: Medium Risks. * Partially: the controls are included only for some components. Furthermore, some controls are not included. ** Limited: the controls are included only for very few components.

6.1.3. Usability Metric

The experts applying FMECA-ATT&CK were given an Excel worksheet with detailed instructions for delivering their input. The experts were instructed to leave a field empty if the task was not clear or they lacked the relevant knowledge needed for delivering a sound judgment. This procedure allows for estimating the usability of the current FMECA-ATT&CK version. In this direction, we define a usability metric to be the ratio of the number of decisions made by an expert to the number of decisions asked to be made by the expert. Table 6 depicts the number of decisions provided to the experts and the number of decisions made; subsequently, used to calculate the usability metric. The experts were given mandatory and optional tasks regarding the risk assessment. The mandatory tasks were system-specific; the expert judgment was expected to be different for different use cases. On the other hand, the optional tasks were non-system-specific, such as the threat checklist, likelihood, and mitigation effectiveness. Furthermore, a decision on an aspect added to the process, not existing in the original proposition is considered optional. Such as the estimation of the environmental and reputation impacts. Offering the experts the option to provide a decision was intended to reduce bias from previous risk assessment processes. Table 6 only depicts the statistics related to the required decisions. The estimated usability of the current FMECA-ATT&CK process was 94.32%. This is an excellent indication of the readiness of the process for application in other use cases. Feedback was received from experts regarding the challenges faced during the execution. The main reason for the lack of ability to provide a judgment was the lack of sufficient background.

**Table 6.** Calculation of the usability metric.

| Use Case Expert | APS | | | DS | | | Usability |
|---|---|---|---|---|---|---|---|
| | 1 | 2 | 3 | 4 | 5 | 6 | |
| Required decisions | 700 | 700 | 700 | 624 | 624 | 624 | |
| Required decisions made | 677 | 608 | 692 | 608 | 538 | 623 | |
| % of required decision made | 96.71% | 86.86% | 98.86% | 97.44% | 86.22% | 99.84% | **94.32%** |

*6.2. FMECA-ATT&CK Questionnaire*

After the execution of the FMECA-ATT&CK risk assessment process, the experts were asked to anonymously answer a questionnaire to rate the method according to the targeted characteristics (Section 5.1.1). The questionnaire is not specific for each of the use cases. Therefore, the compiled results from all the experts are presented in this section. The questionnaire included nine questions, seven of which were related to the targeted characteristics, one regarding the execution time, and the last to record their comments. Additional details regarding the questions are presented in Appendix C. Regarding the execution time, it ranged from 3 to 4 h per expert. The main reason behind this can be linked to the comprehensive nature of the approach which according to the majority of experts was found to be from comprehensive to very comprehensive. The approach was also perceived to be suitable, feasible and highly adaptable for application in several use cases, but requires certain adaptations for its implementation in real systems. The majority of results in the scalability rating suggest that the approach is suitable for implementation in a system of systems with a moderate number of components. Finally, the majority of experts found some of the results to make sense while others did not. This was expected due to the fact that the input used to generate the risk assessment results were consolidated from all the experts in each use case with various diversion in the experts' inputs. Nevertheless, experts' critical comments were received and are presented in Section 6.3 and will be considered for future improvement of FMECA-ATT&CK. Additionally, the risk assessment results when compared to the Bow-Tie results suggest that FMECA-ATT&CK is capable of producing sound results that are comparable with a more granular risk description, adaptable and comprehensive approach.

*6.3. Experts Comments*

The experts were asked to provide their critical comments regarding each step of FMECA-ATT&CK. Several comments were received from different experts. They can be summarized as follows:

- Scope definition (Step 1): The classification criteria for certain components is not clear. Some components can be classified in different ways, others were outside the knowledge field of some experts. Furthermore, additional technical and non-technical components should be considered, such as the human operator. Moreover, there exist several performance standards for defining safety-related failure modes.
- Relevant failure modes (Step 2): The criteria for defining the relevant failure mode was characterized as difficult. Some emphasized existing failure modes that are safety-related are easier to consider than security-related failure modes. Furthermore, human errors were proposed for consideration.
- Impact estimation of failure modes (Step 4): The current estimation criteria are generic and require additional methods such as a hazard and operability study (HAZOP) or event tree analysis (ETA). Furthermore, some failure modes were unclear to some experts and therefore were unable to estimate their impact. Additionally, quantifying the safety, financial, environmental, and reputation criticality scores for certain components was found to be challenging.
- Training: Additional training was required for better execution.
- Scope: Experts with more operational than technical expertise found the approach difficult to apply due to the lack of knowledge of component-level failures. Furthermore, the human element is under-represented in the current approach. Humans can be an asset in the system as well as a risk.
- Background: the approach requires several experts with diverse backgrounds, including operational and technical experts. Some components require specific knowledge to provide a more sound judgment.

## 7. Discussion

This paper presents an empirical study aimed to evaluate the recently proposed FMECA-ATT&CK risk assessment approach for CPSs. The evaluation approach relied on expert judgment. FMECA-ATT&CK was subjected to detailed application and critical comments from a group of experts with various expertises and diverse backgrounds to elicit improvements. In this section, we will summarize the limitations of the FMECA-ATT&CK approach and discuss directions for future work.

Input from experts implementing Bow-Tie referred to the demanding task of conducting a component-level assessment. With the time provided for assessments, only high-level assessment was possible. FMECA-ATT&CK, on the other hand, was originally proposed as a method to reduce the need for expert judgment while at the same time being comprehensive and systematic in its coverage. The expert spent no time identifying threats, estimating their likelihood, or figuring out the required risk controls. Such information was utilized based on the encoded knowledge provided by the ATT&CK framework. The threats were drawn from the list of ATT&CK techniques and their properties. Based on the components' properties in the system model, the relevant ATT&CK techniques are automatically identified as relevant threats. The likelihood values of the ATT&CK techniques are estimated based on the CVSS method and relying on a group of heuristics (more details can be found in [12]). The relevant risk controls for each ATT&CK technique are queried from the ATT&CK repository. Furthermore, the experts were not required to estimate the operational nor staging impact of threats as it was pre-calculated based on a modelled graph of the system. The graph is modelled based on the components' network and application level connections provided as input for the risk assessment. The observed average time for conducting FMECA-ATT&CK was 3 h per expert to provide a comprehensive output. This highlights the utility of FMECA-ATT&CK in achieving its original objective.

However, several aspects were observed when comparing the results obtained through Bow-Tie and FMECA-ATT&CK. One aspect related to the experts' ability to contextualize the system is unmatched in the current form of FMECA-ATT&CK. For instance, in the APS use case, the RCC is expected to be hosted at a university facility. This information is not encoded in the system model. However, it was communicated during the initial session introducing the use case. Although valid accounts are an identified risk by FMECA-ATT&CK, the contextual information that these accounts can be stolen from students is not yet encoded in FMECA-ATT&CK. This affects the communication of the identified risks. Additionally, when discussing possible mitigation methods during the execution of Bow-Tie in the DS use case, the experts suggested future directions that are relevant but not yet implemented, such as zero trust and resilient design. Such strategic directions cannot be made by the FMECA-ATT&CK approach. This sheds additional light on the component level in which FMECA-ATT&CK operates.

### 7.1. Limitations in the Evaluation

We acknowledge the following limitations in the evaluation process:

- The results received from the fourth group might include bias due to the bandwagon effect. This was an unavoidable effect in order to accommodate the participants' time limitations. Efforts to reduce the bias were taken in the form of seeking individual confirmation of the results.
- The FMECA-ATT&CK approach for calculating threat likelihood is based on the calculated CVSS metrics for the techniques in the different ATT&CK matrices which are system-independent and pre-estimated and discussed in previous work [12]. The experts were offered a chance to provide their own estimation but due to time limitations, they were unable to do so. Therefore, we resorted to utilizing the pre-estimated data which is subject to bias.
- We are not claiming that FMECA-ATT&CK is straightforwardly applicable in application domains of CPSs other than maritime and energy. This would require extending the evaluation to include additional and diverse use cases.

### 7.2. Future Work

In summary, based on the results from the evaluation process, the identified future work to improve FMECA-ATT&CK are listed below:

- The scope of considered failure modes focuses on adversarial threats. Considerations of non-adversarial threats, such as human errors, could be useful as a future direction.
- Additional guidelines and supporting methods are needed to estimate the impact of certain failures. Particularly, the estimation of safety and financial impacts.
- The current asset categorization does limit the scope of relevant use cases. Categorizing some components according to the existing asset categorization criteria was found to be challenging. This suggests the proposition of domain-specific categorization. Consequently, the approach requires additional adaptations to accommodate the change of scope. This can include domain-specific threats, failure modes, and risk controls.
- FMECA-ATT&CK is suitable for tier 3 activities according to NIST risk management tiers which address risk from the perspectives of system components [31]. The conducted risk assessment process using Bow-Tie yielded some risk mitigation measures that are at higher tiers, such as a business continuity plan (BCP). The consideration of such mitigation measures requires additional tasks to be conducted after FMECA-ATT&CK which focus on multi-tier risk management rather than tier 3 risk assessment. In this direction, the expansion of the list of supporting controls will be considered in the future.
- The utilization of additional use cases and different application domains for the application of FMECA-ATT&CK will expand its applicability.
- Investigating the efficiency of integrating FMECA-ATT&CK for cyber risk management in real decision-making units (DMUs) would be an interesting direction. For that,

Wang et al. [32] proposed the utilization of data envelopment analysis (DEA) to measure the efficiency of cybersecurity DMUs. This approach would provide quantitative measurements for the reduced cost which FMECA-ATT&CK is hypothesized to achieve as a consequence to reduce the need for expert judgment.

## 8. Conclusions

There is increased interest in cyber–physical systems (CPSs) as their application has been observed in various domains such as energy, manufacturing, and maritime. The cybersecurity aspect of such systems has been the focus of many in academia and industry. In order to improve the risk management capabilities, a number of approaches and methods have been proposed to assess the cyber risks of CPSs. However, there is a lack of dedicated work in the literature that addresses the evaluation of proposed risk assessment approaches. Our evaluation approach in this paper can be useful to evaluate other risk assessment processes. We proposed a set of characteristics to evaluate risk assessment processes and multi-staged execution procedures to measure the process according to a group of characteristics: applicability, feasibility, usability, adaptability, scalability, accuracy, and comprehensiveness. At the same time, reducing the effect of bias introduced by the reliance on experts' judgment was pursued. Recently, a new FMECA-ATT&CK approach has been proposed to address the issue of increased reliance on expert judgment. The approach is based on the failure mode, effects and criticality analysis (FMECA) risk assessment process, enriched with the semantics and encoded knowledge in the ATT&CK framework. FMECA-ATT&CK was subjected to empirical evaluation by applying it to different use cases from different application domains by several groups of experts with various expertise and backgrounds. To provide a comparison basis, Bow-Tie was used as an additional common risk assessment process.

When comparing FMECA-ATT&CK with Bow-Tie for risk assessment, it was found that FMECA-ATT&CK is capable of identifying similar risks, consequences, and risk controls for the same use cases although the assessment was conducted by different groups of experts without any communication between them. This finding highlights the accuracy of the results obtained through the application of FMECA-ATT&CK. Additionally, the comprehensiveness, adaptability, feasibility, and usability of the approach were measured by experts through a questionnaire and were found to be excellent. On the other hand, the scalability was restricted to systems with a moderate number of components. Furthermore, the applicability of the approach was demonstrated through its application in assessing the risks for two CPS use cases in two different application domains, providing logically sound results. In summary, the overall results are positive and suggest that FMECA-ATT&CK is a viable option for design- and component-level cyber risk assessment for CPSs.

However, several areas for improvement have been identified based on experts' input. This includes asset categorization, identification of relevant failure modes, impact estimation, lack of human element, and the scope of the suggested controls. All of these have been discussed in the paper and rendered as suggested directions for future work.

**Author Contributions:** Conceptualization, A.A. and V.G.; methodology, A.A. and V.G.; software, A.A.; validation, A.A. and V.G.; formal analysis, A.A.; investigation, A.A.; resources, A.A. and V.G.; data curation, A.A.; writing—original draft preparation, A.A.; writing—review and editing, A.A. and V.G.; visualization, A.A.; supervision, V.G. All authors have read and agreed to the published version of the manuscript.

**Funding:** This work was funded by the NTNU Digital transformation project Autoferry.

**Institutional Review Board Statement:** Not applicable.

**Informed Consent Statement:** Informed consent was obtained from all subjects involved in the study.

**Data Availability Statement:** Data prepared for the evaluation, results, and utilized templates are maintained in a public repository https://github.com/ahmed-amro/FMECA-ATT-CK-Evaluation (accessed on 28 February 2023).

**Acknowledgments:** The authors would like to express their gratitude to the participating experts, namely, Aida Akbarzadeh, Andre Jung Waltoft-Olsen, Arne Roar Nygård, Erlend Erstad, Filip Holik, Georgios Kavallieratos, Marie Haugli-Sandvik, Mohamed Abomhara, and Vijayan Manogara. The experts' time and efforts invested in this work as well as their valuable comments are highly appreciated and will contribute to advancing the research in the field.

**Conflicts of Interest:** The authors declare no conflict of interest.

## Appendix A. Detailed FMECA-ATT&CK Description

FMECA-ATT&CK approach relies on a group of tables to collect and process the different aspects of risks, namely, threat identification, likelihood estimation, impact estimation, and detectability estimation. Table A1 summarizes the steps of FMECA-ATT&CK, the relevant tables to be filled, the data description, expected values, and the data sources. This highlights areas where experts' judgments were speared. After conducting the steps in the table are completed, a risk priority number (RPN) calculation and mitigation identification (RPNMI) algorithm is executed to generate the results. The algorithm is described in Algorithm A1.

**Table A1.** Detailed description of the FMECA-ATT&CK steps, tables, data types, and knowledge sources.

| Step | Table | Column | Data Description | Data Values | Data Source |
|---|---|---|---|---|---|
| Step 1: Specify Components | Component Description Table (CDT) | Class | Relevant ATT&CK Matrices | Enterprise, ICS, Mobile, Combination | Experts |
| | | Comp Name | Component Name | | Architecture Model |
| | | Type | Component ICS Categorization | Control Server, Data Historian, Engineering Workstation, Field Controller/RTU/PLC/IED, HMI, I/O Server, SIS/Protection Relay, Sensor | Experts choice based on ATT&CK categorization |
| | | Platform | Component IT Platform | Windows, Linux, Network, macOS, Cloud, Containers | Architecture Model |
| | | Technology | Component Technology | App-Based or Other | |
| | | Additions | Component Additions | Radio, GPS, Cell, Wi-Fi, Video, etc. | |
| Step 2: Identify Failure Modes | - | - | Relevant Failure Modes | All ATT&CK Tactics (16) | Experts choice based on ATT&CK Tactics |
| Step 3: Identify Controls | Failure-Mitigation Table (FMT) | Matrix | ATT&CK Matrix | Enterprise, ICS, Mobile | ATT&CK |
| | | Technique | ATT&CK Technique | All ATT&CK Techniques (>700) | |
| | | Mitigation | ATT&CK Mitigation | All ATT&CK Mitigations (>70) | |
| | | Efficiency | Mitigation Efficiency | (0.0–1.0) | Experts |
| | Component-Mitigation Table (CMT) | Mitigation | ATT&CK Mitigation | All ATT&CK Mitigations (>70) | ATT&CK |
| | | Component 1 | Component Name | (0: not covered or 1: covered) | Architecture Model |
| | | Component 2 | | | |
| | | …… | | | Architecture Model |
| | | Component N | | | |
| Step 4: Estimate the Impact of the Consequences of Failure Modes | Failure-Mode-Consequences Table (FMCT) | Matrix | ATT&CK Matrix | Enterprise, ICS, Mobile | ATT&CK |
| | | Tactic | ATT&CK Tactics and Impact Techniques | All Tactics and Impact Techniques (>90) | |
| | | Operational | Wight of Operational Consequence | | |
| | | Safety | Wight of Safety Consequence | | |
| | | Information | Wight of Information Consequence | (0.00–infinity) | Experts |
| | | Financial | Wight of Financial Consequence | | |
| | | Staging | Wight of Staging Consequence | | |

**Table A1.** *Cont.*

| Step | Table | Column | Data Description | Data Values | Data Source |
|---|---|---|---|---|---|
| | | Matrix | ATT&CK Matrix | Enterprise, ICS, Mobile | ATT&CK |
| | | Tactic | ATT&CK Tactics and Impact Techniques | All Tactics and Impact Techniques (>90) | |
| | The Failure-Mode-Metric Table (FMMT) | Operational | Operational Metric to be used | Overall Operational Impact (OOI), Impact to Control Functions (I2CF), Impact to Monitoring Functions (I2MF) | |
| | | Safety | Safety Metric to be used | Safety Criticality (SC) | |
| | | Information | Information Metric to be used | Location Information Criticality (LIC), Information Criticality (IC), Intellectual Property Criticality (IPC) | Process Defined |
| | | Financial | Financial Metric to be used | Financial Criticality (FC), Occurring Financial Criticality (FC2) | |
| | | Staging | Staging Metric to be used | Out-Degree Centrality (ODC), Overall Component Criticality (OCC) | |
| | | Comp Name | Component Name | | |
| | | OOI | OOI score of component | | Graph of Architecture Model |
| | | I2CF | I2CF score of component | | |
| | | I2MF | I2MF score of component | | |
| | | SC | SC score of component | | |
| | Component-Criticality-Scoring Table (CCST) | LIC | LIC score of component | | |
| | | IC | IC score of component | (0.0–1.0) | Experts |
| | | IPC | IPC score of component | | |
| | | FC | FC score of component | | |
| | | FC2 | FC2 score of component | | |
| | | ODC | ODC score of component | | Graph of Architecture Model |
| | | OCC | OCC score of component | | Process Defined |
| | | Matrix | ATT&CK Matrix | Enterprise, ICS, Mobile | |
| | | Technique | ATT&CK Technique | All ATT&CK Techniques (>700) | |
| Step 5: Identify Failure Mechanisms | Techniques-Description Table (TDT) | Tactic | ATT&CK Tactics | All ATT&CK Tactics (16) | ATT&CK |
| | | Platform | Technique IT Platform | Windows, Linux, Network, macOS, Cloud, Containers | |
| | | Type | Technique ICS Assets | Control Server, Data Historian, Engineering Workstation, Field Controller/RTU/PLC/IED, HMI, I/O Server, SIS/Protection Relay, Sensor | |
| | | Technology | Technique Technology | App-Based or Other | Experts |
| | | Additions | Technique Additions | Radio, GPS, Cell, Wi-Fi, Video, etc. | |
| Step 6: Estimate the Likelihood of Failure Mechanisms | Techniques-Description Table (TDT) | CVSS | Technique Exploitability Score based on CVSS | (0.00–3.89) | ATT&CK-based heuristics and Experts |
| Step 7: Evaluate the Risks | - | - | Risk Rating Criteria such as thresholds | e.g., Risk <3 = Low | Experts |
| Step 8: Propose Risk Reduction Measures | - | - | Suggested mitigation methods for each technique | All ATT&CK Mitigations (>70) | ATT&CK |

**Algorithm A1** Risk Priority Number (RPN) Calculation and mitigation identification (RP-NMI) (adapted from [12]). Check Table A1 for acronyms

```
 1: procedure RPNMI(TDT, CDT, FMCT, CCST, FMMT, FMT, CMT)
 2:    for each component in CDT do
 3:        AttackList ← IdentifyRelevantAttacksByMatchingAttributes(CDT, TDT)
 4:        for each attack in AttackList do
 5:            Likelihood ← CalculateAttackLikelihood(CVSSinTDT)
 6:            Impact ← CalculateAttackImpact(RelevantConsequencesinFMCT andmetricsinFMMT
    andcomponentscoresinCCST)
 7:            Detectability ← CalculateAttackDetectability(MaxEfficiencyamongrelevantMitigationsinFMTandCMT)
 8:            RPN ← Likelihood × Impact × Detectability
 9:            MitigationList ← GetAttackMitigation(FMT)
10:        end for
11:    end for
12:    return AttackLists, RPNs and MitigationLists
13: end procedure
```

## Appendix B. Consolidation Process

The consolidation was utilized as a means to implement voting on conflicting decisions in the assessment. Voting applies brainstorming with the nominal group technique. The following protocol was followed for consolidating the results. If a majority is identified for a decision point, the majority decision will be directly used as the input for the assess-

ment. Otherwise, if only a single response is found for a decision point, the response is directly used as the input for the assessment. Conversely, if a response agrees with other non-matching responses, that response is considered inclusive and is used as the input for the assessment. For instance, if the component classification is IT, OT, or IT/OT, then IT/OT is considered inclusive of the other responses. Otherwise, the average is calculated for decisions including numerical values. The conflicting decision points were moved for discussion in stage 3 in the groups. Moreover, an additional step is conducted to rectify any implementation errors. For instance, expert input was considered incorrect under the scope and semantics of the conducted process. For instance, some experts categorized certain components based on their own definition rather than the definition proposed in the process. Additionally, the ratio of consensus is tracked to measure the assessment quality; under the assumption that when a consensus is reached, the input quality for the assessment is higher than in the case of no consensus.

## Appendix C. Questionnaire Details

The experts' feedback was queried through a questionnaire to evaluate FMECA-ATT&CK based on the chosen characteristics. Table A2 depicts the questions sent to the experts and the answer guide.

**Table A2.** Questions and choices used for expert feedback.

| | Question | Choice | Meaning |
|---|---|---|---|
| 1 | How applicable is the approach for application in different CPS use cases? | 1 | Very limited applicable use cases |
| | | 2 | Only few number of applicable use cases |
| | | 3 | Several applicable use cases |
| | | 4 | Many applicable use cases |
| | | 5 | So many applicable use cases |
| 2 | How feasible was the implementation of the different steps? Note: This is related to the approach itself and not the current mode of execution as delivered through the excel sheet | 1 | The entire process is not feasible for implementation |
| | | 2 | Some steps are not feasible for implementation |
| | | 3 | The process is feasible but require some adaptation for implementation |
| | | 4 | The process is feasible and can be implemented in its current form |
| 3 | How reasonable were the results? | 1 | The results did not make sense at all |
| | | 2 | Some of the results did not make sense while others did |
| | | 3 | The results do make sense |
| 4 | How difficult it is to integrate additional aspects? (asset categories, threats, mitigation measures, impact elements, etc.) | 1 | It would be extremely difficult to integrate additional aspects |
| | | 2 | It would require a lot of modifications to integrate additional aspects |
| | | 3 | Integrating additional aspects is possible with minor modifications |
| 5 | How comprehensive is the approach in its inclusion of elements required for sufficient cyber risk assessment processes? | 1 | The approach scope is very limited |
| | | 2 | The approach scope is limited |
| | | 3 | The approach scope is sufficient, but many elements should be added |
| | | 4 | The approach scope is comprehensive; but some elements can be added |
| | | 5 | The approach scope is very comprehensive |
| 6 | How would it perform in large and complex networks or Systems of Systems (SoS)? | 1 | Suitable and efficient only for small SoS |
| | | 2 | Suitable and efficient for moderate SoS |
| | | 3 | Suitable and efficient for large SoS |
| | | 4 | Suitable and efficient for very large SoS |

**Table A2.** *Cont.*

| | Question | Choice | Meaning |
|---|---|---|---|
| 7 | How easy was it to follow with limited training/Consultation? Note: this is related to the current mode of execution as delivered through the excel sheet | 1 | I could not execute the assessment with the amount of training I received. |
| | | 2 | I could only execute some steps of the assessment due to ambiguous tasks. |
| | | 3 | I executed all the required steps but could not finish some of the tasks due to ambiguity |
| | | 4 | I executed all the required steps and finished all the tasks |
| 8 | Would you like to elaborate on the applicability, feasibility, accuracy, adaptability, scalability, and required training to apply the approach? | Open Ended | |
| 9 | How many hours in total did the process took to be completed, approximately. (Filling the Excel sheet) | 1 | an hour or less |
| | | 2 | around 2 h |
| | | 3 | around 3 h |
| | | 4 | around 4 h |
| | | 5 | 5 h or more |

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
