# Peer review of "Evaluation of a Cyber Risk Assessment Approach for Cyber–Physical Systems: Maritime- and Energy-Use Cases"

_jmse, doi:10.3390/jmse11040744_

Round 1
Reviewer 1 Report (Previous Reviewer 3)
Please read the attachment. Thank you.

Author Response
The authors would like to ask the reviewer for his valuable comments. Our responses to the comments are detailed in the attached document.

Reviewer 2 Report (New Reviewer)
The authors presented and empirically conducted a standard-aligned evaluation methodology to evaluate a newly proposed cyber risk assessment approach for cyber-physical systems (CPS). The approach is called FMECA-Aat&CK. The results highlight the positive utility of FMECA-ATT&CK in model-based, design-level, and component-level cyber risk assessment for CPS with several identified directions for improvements. Moreover, the standard-aligned evaluation method and the evaluation characteristics have been demonstrated as enablers for the thorough evaluation of cyber risk assessment methods.
line 147: Then, The … -> Then, the …
line 180: measures. to identify gaps. -> measures to identify gaps.
Author Response
Thank you for your comments. The found mistakes have been rectified.
This manuscript is a resubmission of an earlier submission. The following is a list of the peer review reports and author responses from that submission.
Round 1
Reviewer 1 Report
Review report of article jmse-2104975
In brief, the paper titled “Evaluation of a Cyber Risk Assessment Approach for Cyber-Physical Systems: Maritime and Energy Use Cases”
(a) is an important and exciting work;
(b) proposes a novel assessment approach for cyber risk assessment;
(c) is based on an innovative combination of different well known methods;
(d) applies the selected methodological approaches quite solidly;
(e) fits the journal’s scope and standards.
General findings related to the manuscript:
The work can significantly impact the cyber security research community and society since the proposed method has a logical process and a firm practical basis. Let the reviewer say it was good to read this work and its “predecessor study” referenced [12] in the manuscript text. After some modifications, the reviewer is sure the manuscript will be worth publishing in JMSE. Although the practical aspect of the manuscript is refreshingly significant, there are some comments from the perspective of decision sciences and practice. However, some major structural suggestions are also made.
Based on the previously described, the manuscript is worth publishing in Journal of Marine Science and Engineering after major modifications.
The detailed comments following the structure of the paper are presented in the followings:
1.
Introduction section
The selection of FMECA for combining with ATT&CK is so direct for a scientific paper. The reviewer understands that ATT&CK is essential in the combination, but FMECA is not. There are many other options, for example, almost any of the MCDM (Multi-Criteria Decision-Making ) methods, or similar methods to FMECA like PRISM (Partial Risk Map) method or traditional FMEA, etc. However, FMECA, FMEA, and PRISM methods are all (very limited) MCDM methods on the field of risk assessment. Cyber security analysis is a subject of complex and multi-attribute assessment. Please review and refer to at least the following works and provide arguments in the introduction section, why FMECA was applied for the combination with ATT&CK (There can be many objective reasons, please refer to some.)
https://doi.org/10.1201/b11032
https://doi.org/10.1016/S0377-2217(03)00020-1
https://doi.org/10.1016/0270-0255(87)90473-8
https://doi.org/10.3390/fi12110205
https://doi.org/10.1016/j.promfg.2020.01.077
https://doi.org/10.3390/math10050676
https://doi.org/10.12700/APH.18.7.2021.7.5
2.
Methodology Section and Evaluation Results Section
Please restructure this part of the manuscript since the methodology, the materials and the results should be interpreted individually. An example: in the scientific methodology section, there is no space for introducing the expert characteristics since it is the subject of the practical case studies. So please focus on only the theoretical process in the methodology section all the practical elements should be removed after the Methodology section. (From the reviewer’s perspective, this work is a typical “Method and Case study” article, where the results can be introduced in the case study section.)
3.
Please provide a more in-depth analytical introduction of the FMECA and Bow-Tie methods to create a transparent and clear background for those who want to apply this proposed methodological combination in the future. How is RPN calculated, what scales were applied for the assessment, etc.
4.
Please provide at least a brief overview of the quantitative results of the assessments.
5.
Discussion section
The discussion section is almost only about the interpretation of the results. The proposed methodology or the case studies’ results are weakly put in the context of other published papers. The proposed assessment process is exciting and promising. Thus, there is the opportunity to compare it with any other techniques or discuss the possible advantages or disadvantages of replacing some methods with others in the proposed process. (For example the replacement of FMECA with PRISM, AHP, TOPSIS, VIKOR, etc. methods.) In general, the combination of risk assessment techniques with any well-known classic MCDM methods are hot scientific topic. Anyway, please put this study into the context of more relevant articles.
6.
Future research directions
Please focus all future research directions on the discussion section. (Some are placed in the results section, no need for them there.) An idea: since the authors identified many future research options, they could be organized in a sub-section of the discussion section.
Overall proposal to the Editor:
The manuscript has significant potential, and the reviewer suggests publishing the manuscript after major changes are made based on the comments above.
Author Response
Response to reviewer's comments are attached

Reviewer 2 Report
Since, as per the authors, their proposed approach is conceptualized as a system, the evaluation phase should also include using Third-Party Risk Management (TPRM) scores as their baseline/references.
Since they are merely using use cases, the scores provided by third-party external vendors, should they suggest congruency to this paper’s findings, will affirm the external validity of this work. Please refer to Keskin et al. (2021)’s work on TPRM scoring (these are generic risk scores provided on usual specifications provided by experts and should also apply to this work). Several of these vendors perform non-intrusive free tests/evaluations of systems, including the application domains of this paper.
Author Response
Responses to the reviewer's comments are attached

Reviewer 3 Report
Please read the attachment. Thank you.

Author Response

(The authors gave the same response as above.)

Round 2
Reviewer 1 Report
The paper is developed and the reviewer understands the opinions of the author. The answers of the author are accepted, and the reviewer wishes the author many more well-established researches in the future.
The reviewer suggests publishing the manuscript.
Author Response
The authors would like to thank the reviewer for his valuable feedback
Reviewer 2 Report
No changes/revisions at all were made.
Author Response
The reviewer merely suggested the utilization of an external paid service which we found non-essential for the scope of this article. No other comments were provided by the reviewer.